# Antioxidant Protection against Trastuzumab Cardiotoxicity in Breast Cancer Therapy

**DOI:** 10.3390/antiox12020457

**Published:** 2023-02-10

**Authors:** Gabriel Méndez-Valdés, Francisca Gómez-Hevia, Maria Chiara Bragato, José Lillo-Moya, Catalina Rojas-Solé, Luciano Saso, Ramón Rodrigo

**Affiliations:** 1Molecular and Clinical Pharmacology Program, Institute of Biomedical Sciences, Faculty of Medicine, University of Chile, Santiago 8380000, Chile; 2Department of Biomedical Sciences, Humanitas University, 20090 Milan, Italy; 3Department of Physiology and Pharmacology “Vittorio Erspamer”, Faculty of Pharmacy and Medicine, Sapienza University, P.Le Aldo Moro 5, 00185 Rome, Italy

**Keywords:** antioxidants, cardiotoxicity, immunotherapy, oxidative stress, trastuzumab

## Abstract

Breast cancer is the most frequent malignant neoplastic disease in women, with an estimated 2.3 million cases in 2020 worldwide. Its treatment depends on characteristics of the patient and the tumor. In the latter, characteristics include cell type and morphology, anatomical location, and immunophenotype. Concerning this latter aspect, the overexpression of the HER2 receptor, expressed in 15–25% of tumors, is associated with greater aggressiveness and worse prognosis. In recent times some monoclonal antibodies have been developed in order to target HER2 receptor overexpression. Trastuzumab is part of the monoclonal antibodies used as targeted therapy against HER2 receptor, whose major problem is its cardiac safety profile, where it has been associated with cardiotoxicity. The appearance of cardiotoxicity is an indication to stop therapy. Although the pathophysiological mechanism is poorly known, evidence indicates that oxidative stress plays a fundamental role causing DNA damage, increased cytosolic and mitochondrial ROS production, changes in mitochondrial membrane potential, intracellular calcium dysregulation, and the consequent cell death through different pathways. The aim of this review was to explore the use of antioxidants as adjuvant therapy to trastuzumab to prevent its cardiac toxicity, thus leading to ameliorate its safety profile in its administration.

## 1. Introduction

Cancer ranks as one of the leading causes of death worldwide and is an important barrier to increasing life expectancy [1]. According to the World Health Organization (WHO) in 2019, cancer was the first or second cause of death before 70 years old in 112 of 183 countries [2]. At present, breast cancer (BC) is the most common cancer in women, having an important negative impact on public health [3]. In 2020, there were 2.3 million women diagnosed with BC and 685,000 deaths globally [4], being the leading cause of mortality among females [5]. At the end of 2020, there were 7.8 million women alive who were diagnosed with BC in the past five years, making it the most prevalent cancer in the world [4], representing 11.7% of total cases of cancer [6]. In more developed regions, overall, 5-year survival of BC is over 80%, in comparison to less developed countries such as India, where the 5-year survival is less than 70%, and South Africa, where it is less than 50% [6].

Breast cancer is a heterogeneous, phenotypically diverse disease with several subtypes with different behavior between them [7]. Its diagnosis is made by biopsy. The most common histologic types are the invasive ductal carcinoma (50–70% of patients), followed by invasive lobular carcinoma (5–15% of patients) and mixed ductal/lobular carcinomas and other histologies in the rest of the patients [8]. Based on molecular and histological evidence, BC is categorized into three groups: BC expressing hormone receptor (HR), which refers to the estrogen receptor (ER) and progesterone receptor (PR), BC expressing human epidermal receptor 2 (HER2) and triple negative BC (ER-, PR- and HER2-) [9]. Overexpression or the amplification of HER2 oncogene, leading to overexpression of this transmembrane tyrosine kinase receptor, is present in approximately 15–25% of all breast cancers, and it is more likely to be diagnosed in younger patients and at a more advanced stage [7,9,10,11]. The HER2 overexpression in primary tumor tissue is associated with a worse prognosis in untreated patients with breast cancer [9,12].

Breast cancer treatment should be based on its molecular characteristics [9]. The most common BC type is the one that expresses hormone receptors, representing 60–70% of all kinds of BC in premenopausal women from developed countries [13].

The most used treatments for this type of BC are tamoxifen as an estrogen blocker and aromatase inhibitors such as letrozole and anastrozole [14]. Patients with amplified or overexpressed HER2 (HER2+) benefit from HER2 targeted therapy, including HER2 antibodies such as trastuzumab (TZB) and pertuzumab and small-molecule tyrosine kinase inhibitors such as lapatinib and neratinib [8]. The standard treatment for BC patients HER2+ is a HER2 antibody plus chemotherapy [15]. The development of TZB has changed HER2-positive BC from an aggressive disease to one with a relatively favorable prognosis [16]. This drug is not exempt of adverse effects, with a vast variety of manifestations [17], where the most frequent are decreased left ventricular ejection fraction (LVEF), infections, abdominal pain, chills, pain, vomiting, headache, and skin rash [18,19,20,21]. The study of cardiovascular adverse events, such as a decrease of LVEF with or without symptoms, and symptomatic heart failure [22,23] are of great interest. The definition of cardiotoxicity using LVEF is not standardized between studies [24]. The mechanism of this adverse event is not clear, but evidence shows that oxidative stress plays an important role [25]. In general, the antioxidant system corresponds to a set of molecules and mechanisms, which have an extremely important role for the cell, because it is responsible for maintaining oxidative levels in a stable state, especially during the increase in the production of free radicals to which cells are exposed. This system is composed of both enzymatic and non-enzymatic antioxidants, with catalase (CAT), superoxide dismutase (SOD), peroxiredoxins, thioredoxins, glutaredoxins, and glutathione peroxidase (GPX), being the main enzymes and the first line of defense against oxidative damage. On the other hand, non-enzymatic antioxidants include reduced glutathione (GSH), nicotinamide adenine dinucleotide phosphate (NADPH), and exogenous molecules including vitamin C and E, carotenoids, flavonoids, and polyphenols, among others [26]. Therefore, this review investigated the use of antioxidants as an adjuvant therapy to TZB, potentially decreasing its cardiotoxic effects.

## 2. Trastuzumab Therapy in Breast Cancer

Human epidermal growth factor receptor 2 is part of a bigger family of epidermal growth factor receptors that comprises HER1, HER2, HER3, and HER4 [27]. In particular, HER2 is of crucial importance, since its overexpression triggers multiple downstream pathways required for the abnormal proliferation of cancer cells. Typically, HER2 is expressed at a low level on the surface of epithelial cells, and it is necessary for the normal development of many tissues, including those of the breast, ovary, lung, liver, kidney, and central nervous system. In contrast, in breast cancer cells, immunohistochemical analyses have revealed extremely high levels of HER2, which can reach up to two million receptors per cell [28].

Physiologically, HER2 activation happens through a receptor homodimerization or heterodimerization. Its activation then triggers a broad spectrum of downstream cascades to promote numerous outcomes, including cell growth, proliferation, and survival. The phosphorylated tyrosine residues on the intracellular domain activates phosphoinositide 3-kinase/protein kinase B (PI3K/AKT), which drives cell survival. Also, the mammalian homologue of the son of sevenless (SOS) activates via MAPK, driving cellular proliferation. One of the downstream effects is the production of vascular endothelial growth factor (VEGF) supporting angiogenesis [27].

Trastuzumab is a monoclonal antibody against HER2 [22]. It consists of two antigen-specific sites that bind to the juxtamembrane of the extracellular domain of HER2, thus preventing the activation of its intracellular tyrosine kinase receptor [27]. Based on that, the mechanisms of action of TZB can be divided into three main subgroups: HER2 degradation, antibody-dependent cytotoxicity (ADCC), and MAPK and PI3K/AKT interference. While the latter mechanism is the most well-known, the former is still concerning. Indeed, it was observed that the binding of TZB to HER2 recruits tyrosine kinase–ubiquitin ligase c-Cbl to its docking site, where HER2 degradation happens [27].

Regarding the antibody-dependent cytotoxicity mechanisms, it has been demonstrated by Clynes et al. that natural killer cells could target HER2-overexpressing cells coated with TZB via a CD16-mediated ADCC mechanism [29].

Treatment with TZB has shown a lot of benefits in patients with BC in terms of survival, reduced recurrences, metastases rates, and second tumors other than BC [30]. A four-year follow-up randomized controlled trial showed that treatment with adjuvant TZB for 1 year is associated with persisting benefits in women with early HER2+ BC [31]. An 11-year follow-up of TZB after adjuvant chemotherapy in patients with HER2+ early BC showed that 1 year of adjuvant TZB significantly improves long-term disease-free survival, but 2 years of TZB had no additional benefit [32].


*Trastuzumab and Cardiotoxicity*


Despite the demonstration that TZB improves survival in non-metastatic breast cancer patients, its mechanism of action brings about a series of adverse effects among which cardiotoxicity has been shown to be the most important. This is because TZB also blocks the function of neuregulin, which is required for normal cardiac growth and maintenance [16].

In fact, TZB is crucial in the prognosis of HER2+ subtypes of cancer patients; approximately 3 to 7% of patients receiving this drug as monotherapy have experienced cardiac dysfunction [33]. Trastuzumab is used frequently with anthracyclines, and its cardiotoxicity is not dose-dependent and is usually reversible, unlike anthracycline [16]. However, it has been questioned whether TZB cardiotoxicity is always reversible [34]. In this combination, it has been reported that this adverse effect increases up to 13% in patients using paclitaxel with TZB and 27% if used in combination with an anthracycline [33]. In a Phase 3 clinical trial developed by Slamon et al. [10], patients with HER2+ metastatic breast cancer, 27% of the subjects receiving TZB plus chemotherapy experienced cardiac dysfunction, compared with 8% of patients receiving chemotherapy alone. Gianni et al. in a RCT with treatment with TZB for 1 year after adjuvant chemotherapy reported 87 patients (5%) where TZB was discontinued due to cardiac problems, with 33 patients with symptomatic congestive heart failure and 62 with confirmed significant left ventricular ejection fraction (LVEF) drop [31].

Chen et al. identified 45.537 old women (mean age 76.2 years) with early-stage BC. Compared with patients who did not receive adjuvant chemotherapy or TZB, the use of TZB was associated with an absolute 14% higher adjusted incidence rate for heart failure (HF) or cardiomyopathy (CM) over 3 years, and patients who received TZB plus anthracyclines had an absolute 23.8% higher rate. Patients treated with anthracycline chemotherapy alone had an absolute 2.1% higher rate of HF or CM over 3 years [35].

Different interventions have been tested in an attempt to show that non-pharmacological measures such as regular, moderate intensity, supervised exercise could function as a primary prevention measure against heart failure in this group of patients [36]. The most used drugs have been beta-blockers, angiotensin-converting enzyme inhibitors, and angiotensin II receptor antagonists. Among beta-blockers, carvedilol [37] and bisoprolol [38] have been shown to have a protective effect against TZB cardiotoxicity. In the same context, lisinopril [37] and perindopril [38] have been tested, showing a similar effect to previous drugs. These findings could be more beneficial if initiated in the first 6 months post-treatment [39]. On the other hand, candesartan has been shown to have beneficial effects in mitigating LVEF decline [40], but a study by Boekhut et al. [41] found no benefit with this drug. Other interventions tested have been the use of statins, where Calvillo-Argüelles [42] found that this treatment was independently associated with a lower risk of cardiotoxicity, while in a study by Abdel-Qadir et al. [43], no statistical association was found. On the other hand, eplerenone has been tried, which has also been ineffective [44]. Other drugs such as sacubitril/valsartan are currently being tested [45]. The results obtained, and the discrepancies may be due to the different inclusion criteria, endpoints, chemotherapy, and immunotherapy regimens.

## 3. Oxidative Stress

As human beings, we function through aerobic conditions, which implies being exposed to different oxidizing compounds such as reactive oxygen species (ROS) and nitrogen species (RNS). Both play essential roles in normal cell function at low concentrations. However, an imbalance has significant negative and irreversible consequences, damaging macromolecules such as proteins, lipids, and nucleic acids [46]. Both ROS and RNS have radical and non-radical species. The first group contains the superoxide anion (O_2_^•−^) and the hydroxyl (•OH) and nitric oxide (NO^•^) radicals. On the other hand, the species that are not radicals are singlet oxygen (^1^O_2_), hydrogen peroxide (H_2_O_2_), hypochlorous acid (HClO), and peroxynitrite anion (ONOO^−^) [47].

At the cell level, there are different ROS sources, among which, the main ones are mitochondria, endoplasmic reticulum, and peroxisomes [48]. There are also ROS-generating enzymes such as NADPH oxidase, xanthine oxidase, and nitric oxide synthase. However, there are also exogenous factors, such as alcohol, tobacco, ultraviolet light, pollution, industrial solvents, and drugs, that form oxidative molecules [49].

### 3.1. Oxidative Stress in Cancer

Regarding oxidative stress in cancer, there is an inevitable vicious circle between its pathogenesis and its treatment. Oxidative stress is involved in many physiological and pathophysiological processes. Cancer is a disease that itself produces oxidative stress and also decreases the enzymatic activity of antioxidant systems such as GPX, SOD, CAT, and non enzymatic molecules, such as vitamin C and E, among others [50].

Higher or lower ROS levels can modify the survival rate of cancer cells through oncogenic signaling, metabolic activity, mismatch of electron transport chain (ETC), inflammation, and hypoxia [51]. Cancer cells are characterized by needing high amounts of adenosine triphosphate (ATP), and the mitochondria is an essential organelle. Mitochondrial ROS play an essential role in causing DNA changes, amplifying and worsening neoplastic transformation. In addition, it increases the survival of these same cells [48].

During cancer development, an interplay between oxidative stress and anticancer immunity tumor exists. Tumoral cells can express suppressor cytokines (IL-6, IL-10, TFGβ, TNFα), which promote tumor growth and metastasis [52], suppressor cells (LT reg), and immunosuppressive molecules. These responses are closely linked to the production of oxidative stress and the generation of ROS, since the processes are regulated by the release, presentation, and regulation of tumor antigens, the stimulation of effector immune cells, and the inhibition of immune regulatory cells [53,54]. In addition, tumor cells preferentially use the glycolysis pathway for ATP synthesis (Warburg effect), which is less effective than the mitochondrial chain, even in the presence of oxygen [55].

### 3.2. Oxidative Stress and Trastuzumab

Although HER2 is overexpressed in a percentage of breast cancers and TZB is used under HER2 (+) conditions, the cardiotoxicity of this drug questions its safety, and the mechanism is not entirely clear. A recent study showed that DNA damage and ferroptosis are implicated in anti-tumor drug cardiotoxicity as well as increased heart weight, interstitial fibrosis, contractile dysfunction, and oxidative stress [56]. Likewise, preventive interventions are not sufficiently developed to be applied in a clinical setting. Because of this, antioxidants and other pharmacotherapeutic options such as angiotensin converting enzyme inhibitor, beta-adrenergic blocking agents, and SGLT2 inhibitors play a key role in reducing cardiotoxicity induced by oxidative stress and other mechanisms [56,57,58]. There are many pathways whereby this damage occurs, such as the inhibition of the mitogen-activated protein kinase/extracellular signal-regulated kinase (MEK/ERK), PI3K/AKT and rat sarcoma protein/mitogen-activated protein kinase (RAS/MAPK) pathways, inducing the production of ROS, mitochondrial impairment, changes in normal cell function, and cell death. Indeed, it seems that trastuzumab leads to an overall increase of all types of ROS. There are several studies supporting this assertion, indeed, Gordon et al. claimed that erbB2 blockade leads to activation of a mitochondrial pathway that results in an increase in cellular ROS production and subsequent cell death [59].

Human epidermal receptor 2 signaling activates prosurvival pathways, maintaining ATP levels and keeping ROS levels at a low concentration [60]. Gordon et al. showed a significant dose-dependent increase of ROS production with HER2 antibody treatment in rat cardiomyocytes [59].

Neuregulin-1 (NRG1) signaling interferes with oxidative stress that activates HER4, which forms heterodimer with HER2 to initiate cardiomyocyte survival and myofibrillar homeostasis pathways. When TZB binds to HER2, it compromises its ability to form homo- or heterodimer with other receptors, affecting the cardioprotective properties of NRG1 and causing cardiac dysfunction [60]. NRG1 activates PI3K/AKT and MAPK/ERK1/2 through HER2 phosphorylation. The deficiency of this pathway increases the susceptibility to heart failure, affects inotropic effects and decreases eNOS, and increases iNOS in cardiomyocytes, further increasing ROS production. Moreover, inhibition of NGR1/HER2 also affects RAS/MAPK (Figure 1) [61,62]. The interruption of all these pathways can be a basis for the application of a combined pharmacological and antioxidant therapy.

The binding of TZB to HER2 induces a change in pro- and anti-apoptotic Bcl-XS/Bcl-XL ratio (Figure 1), decreasing Bcl-XL (antiapoptotic) and increasing Bcl-XS (proapoptotic). Both are proteins correlated with the correct mitochondrial function and apoptosis. The shift in this balance leads to cell death through the caspase 3/9 stimulation [63,64]. As a consequence, there is a malfunctioning of ETC, generation of free radicals, release of pro-apoptotic proteins, and the uncoupling of oxidative phosphorylation. Since cardiomyocytes require high energy to function, the mitochondrial dysfunction and decrease of ATP result in impaired cardiac function [65].

Trastuzumab-mediated HER2 signaling dysregulation dampens the MEK/ERK pathway (Figure 1), which induces apoptosis and changes in normal cellular function. In addition, it causes a disruption in mitochondrial permeability transition pores (mPTP), increasing Ca^2+^ overload, further ROS production, and cardiac tissue injury. These actions end in a decrease of cell survival and proliferation [66,67].

PI3K/AKT/mTOR pathway is a key factor in HER2 signaling (Figure 1). Protein kinase B, as well as the pathway mentioned above, cell survival, growth, and its cycle. The inhibition of this cascade results in autophagy inhibition, causing a massive accumulation of damaged mitochondria and free radicals, producing oxidative stress and toxicity in cardiomyocytes [60,68,69].

In cancer, the excessive expression of HER2 causes an upregulation of MAPK playing a key role in the cancer development and being related with the production of cardiac impairment and cardiotoxicity. Mitogen-activated protein kinase modulates the cell homeostasis through the regulation of the proliferation, differentiation, survival, development, stress response, and apoptosis [70]. Cancerous mutations in MAPK pathways are related with Ras and B-Raf in the extracellular signaling. There are differences at the time of the integration of the signals that widely vary according to the tumor type but are important for the monitoring of outcome and the response to the treatment. On the other hand, MAPK repeals antiestrogen resistance in human breast cancer [71,72].

## 4. Potential Role of Antioxidants as Adjunct Therapeutic Agents

As previously mentioned, the relationship between cancer and oxidative stress has been demonstrated via elevated levels of oxidative stress biomarkers and decreased levels of antioxidants in cancer patients [50]. However, the role of oxidative stress is variable between the different types of existing cancer and the signaling pathways involved [73]. In the case of breast cancer, it has been reported that types positive for estrogen receptors present high levels of oxidative stress added to the fact that those triple negative breast cancer would present lower levels of 8-hydroxy-2′deoxyguanosine, thus supporting the role that estrogen receptors such as HER2 could have in the induction of oxidative stress in breast cancer and even indicating a possible differentiation between the various breast cancer subtypes and the role of oxidative stress in each of them [73,74,75,76,77]. In addition, the levels of oxidative stress would be related to the therapy used, where the role of oxidative stress induced by anthracyclines within their mechanism of action has been very well-documented, which entails a high risk of cardiotoxicity in those patients [78]. Trastuzumab as immunotherapy in breast cancer also induces early changes in the levels of biomarkers of oxidative stress and antioxidants, having been reported to have increased malondialdehyde (MDA) levels and decreased SOD activity in patients who received TZB compared to the levels found prior to the treatment. It has also been reported that the decrease in LVEF would correlate negatively with changes in MDA levels and positively with changes in SOD activity [25]. For this reason, the use of antioxidants has been proposed to reduce the oxidative stress concomitant with the use of trastuzumab and thereby prevent the cardiotoxic effects. Various preclinical studies have tested this idea, their results being summarized in Table 1.

### 4.1. Nuclear Factor Erythroid-2-Related Factor 2

As previously mentioned, using oxidative stress as a target to lessen cardiotoxicity induced by cancer therapy is plausible, and numerous adjuvant drugs have been tested to attenuate this cardiac damage, where the redox state can be modified by supplementing exogenous antioxidants. In addition, it could resort to the use of drugs that activate endogenous antioxidant response pathways, such as nuclear factor erythroid-2-related factor 2 (Nrf2). However, this last point gives a space for discussion in the literature because of the dual role that the Nrf2, the main regulator of the antioxidant response, could have in the context of cancer. It is considered an indirect antioxidant and is released from its KEAP1 repressor protein when there is an increase in oxidative stimulus under certain ranges, being able to translocate to the nucleus and bind to the endogenous antioxidant response site (ARE), starting transcription of genes encoding for both enzymatic and non-enzymatic antioxidants molecules, such as SOD, CAT, GSH, GPX, GR, among others [73,93,94]. However, in a normal cell context, Nrf2 has a preventive role on carcinogenesis by reducing oxidative stress and promoting gene repair. In addition, the prolonged activation of this transcription factor could be deleterious in the context of cancer cells, since it could finally deliver mechanisms of resistance to therapy, evading cell death and promoting malignancy and cancer metastasis [95].

### 4.2. Dexrazoxane

Currently, dexrazoxane (iron chelator, acting indirectly as an antioxidant by preventing the formation of hydroxyl radical) is the only medication approved by the United States Food and Drug Administration to prevent anthracycline cardiotoxicity; however, its use is highly restricted to patients with metastatic breast cancer who have received a cumulative dose greater than 300 mg/m^2^ doxorubicin and need to continue therapy to maintain tumor control [78]. Therefore, there is already previous data for the appropriate use of an antioxidant as adjuvant therapy to attenuate cardiotoxicity induced by cancer therapy.

### 4.3. Monoclonal Antibodies and N-Acetylcysteine

In the case of specific harm caused by TZB, a preclinical study evaluated the role of oxidative stress in the myocardial damage produced by the blockade of erbB2 receptors through the use of monoclonal antibodies in a culture of rat cardiomyocytes, reporting that the cardiomyocyte death would occur through a ROS-mediated mitochondrial pathway. In turn, the study also demonstrated that the use of N-acetylcysteine ameliorated the damage, so the authors concluded that the regulation of the redox state could be an option to attenuate the cardiotoxicity induced by TZB [59].

### 4.4. Polyunsaturated Fatty Acids

Polyunsaturated fatty acids (PUFAs) are fatty acids that have more than one double bond between their carbons and this group includes omega-3 and omega-6 fats. Both are essential for human beings since they are not produced endogenously, but we must obtain them from food. PUFAs in recent studies have shown pleiotropic actions on cell function, such as their antioxidant effect, since they are incorporated into the cell membrane and are capable of modifying redox signaling. This is related to the prevention of cardiovascular diseases [96,97], especially with omega-3 due to the reduction of risk factors and hyperlipidemia, inhibition of endothelial dysfunction, inhibition of inflammation, reduction of oxidative stress, antiarrhythmic effects, vasodilation, and reduced blood pressure [98,99]. Moreover, Abdellatif et al. showed that calanus oil (rich in omega 3) has the potential of regulating myocardial remodeling and oxidative stress, resulting in a anti-hypertrophic effect. This study also demonstrated that this PUFA decreased the elevated cardiac enzymes (LDH and CK-MB) and MDA, increased antioxidant status of the heart, and ameliorated the histopathological and structural changes in cardiac tissues and prevented myocardial fibrosis [100].

However, there are debatable outcomes between different studies according to the methodology used and the clinical stage of the disease [101,102]. Regarding the PUFA prevention of cardiotoxicity induced by antitumoral agents such as TZB, there is a lack of information about it, and most of the investigations have been carried out under the conditions of chemotherapy and radiotherapy or adjuvant therapies. Nevertheless, studies showed that PUFA (omega 3) may attenuate the side effects of the therapy and maintain homeostasis and reduce oxidation and its consequent inflammation, which induces accumulation of dysfunctional proteins and DNA damage in muscle as well as preservation of muscle and the increase or maintenance of body weight and improves treatment tolerance [103,104]. In addition, TZB reduces breast tumor growth and enhances the effectiveness of the treatment, reducing HER2 signaling in vitro [81]. Likewise, Ravacci et al. showed that PUFAs modify HER-2 signaling, causing a disrupted lipid raft through decreasing activation of AKT, ERK1/2, and induced apoptosis [105].

### 4.5. Coenzyme Q10

A study carried out in a culture of human fetal cardiomyocytes demonstrated that the use of coenzyme Q10 increased cell viability and decreased biomarkers of oxidative stress and inflammation [83]. In vivo studies carried out in rats to evaluate cellular response after being subjected to TZB plus antioxidant treatment have also obtained favorable results, reporting increased cell viability or decreased apoptosis in rat cardiomyocytes [80,85,86,91,92].

### 4.6. Ascorbic Acid

Ascorbic acid or vitamin C is an organic acid with antioxidant properties. Humans and other animals are not capable of synthesizing it endogenously, therefore it must be ingested through food. In one study, high doses of AA (≥10 mM) significantly reduced cell viability of all breast cancer cell lines in combination with conventional antitumor drugs such as TZB, eribulin mesylate, tamoxifen, or fulvestrant [79], and good tolerance and safety have been demonstrated. It has been suggested that the anticancer effect of AA is through H_2_O_2_ accumulation due to CAT deficiency in cancer cells and subsequent cell death by apoptosis, pyknosis, and necrosis and through the nuclear translocation of an apoptosis-inducing factor. Therefore, millimolar concentrations of extracellular AA kill cancer cells but not normal cells [106,107]. Although the use of AA in an antitumoral context is still controversial, this is a mainstay of breast cancer treatment when compared to chemotherapy or endocrine therapy alone [79].

### 4.7. Other Antioxidants Molecules with Antioxidant Effect

Other studies that have also focused on studying mitochondrial dysfunction due to damage by TZB in rat hearts through the use of antioxidants or drugs with antioxidant effects, such as curcumin, chrysin, thymoquinone, melatonin, metformin, α-linolenic acid, flaxseed, and secoisolariciresinol diglucoside have obtained favorable results, improving mitochondrial dysfunction and being associated with an increase in antioxidant enzymes and decreased oxidative stress biomarkers [82,85,87,89].

Most of these studies are described in Table 1. Therefore, given these findings, there is a need to continue carrying out preclinical studies and advancing existing clinical studies that can test antioxidant compounds as pharmacological agents for an adjuvant therapy to prevent cardiotoxicity mediated by oxidative stress induced by TZB.

## 5. Discussion

Currently, BC continues to be a major public health problem worldwide, having major repercussions on both survival and quality of life. Therefore, it is important to assess the effectiveness of its current therapies to reduce its burden on the general population. Multiple clinical trials have shown that HER2-targeted treatment with monoclonal antibodies such as TZB in patients with HER2+ breast cancer have advanced survival outlook, especially considering that these patients started with a very poor prognosis. However, the therapeutic efficacy of TZB varies among patients [108,109,110] and the tumor microenvironment [53]. Numerous studies have undeniably confirmed that its pharmacological beneficial effect is also accompanied by the occurrence of oxidative stress in cardiomyocytes, which is at the base of the pathogenic mechanisms of the cardiotoxicity caused by this drug. Patients’ risk of developing this adverse effect varies depending on their characteristics. The elderly group of patients represent a significant yet increasing percentage of patients with breast cancer, and, due to their age, they have an increased cardiovascular risk [35]. Because of this, cardiotoxicity of TZB should be further investigated especially in this group.

Available data concerning cardiotoxicity suggests the employment of an adjuvant therapy aimed to reinforce the antioxidant defense system, achieving attenuation of this adverse effect. So far, clinical studies considering this potentially protective effect in this scenario have not been performed. Antioxidants have been associated in promising studies with decreased recurrence of BC [111] and could have a decreased risk of mortality [112] or not have a significant effect [113]. This is probably because antioxidant supplements and drugs are a vast group that have not been properly characterized with respect to their pharmacological parameters and possible interactions with other drugs, so it is unknown whether any intervention could be detrimental to the patient’s current therapy.

Nevertheless, it is worth considering the efficacy of antioxidants both in TZB as monotherapy or in combination with other chemotherapeutic agents, which have been successfully proven in preclinical studies with different models.

Considering the benefits of antioxidants in experimental models, there is an unresolved need to conducted clinical studies on the effectiveness and safety of an antioxidant booster in patients on TZB therapy. Such studies may help in reducing or preventing TZB cardiotoxicity, thus avoiding the withdrawal of this drug from the patient’s treatment and therefore improving the clinical outcome of patients.

Therefore, in this review, we exhibited the association that has been made between TZB and an augmented oxidative stress, as well as the existence of successful preclinical studies on antioxidants as protective agents against TZB-induced cardiotoxicity with the aim of designing future clinical research on the subject.

## 6. Conclusions

The addition of TZB to the treatment of BC has shown great progress in terms of survival; however, it still has limitations, mainly in its most important drug-related toxicity, the TZB-related cardiotoxicity. Based on the evidence presented and the proposed relationship of this adverse event with oxidative stress, pursuing studies on this topic would aid in preventing or attenuating the onset of cardiotoxicity, guaranteeing an uninterrupted treatment to patients receiving this therapy. Moreover, the antioxidant therapy, in addition to having a cardioprotective effect, could prove to directly benefit cancer treatment.

## Figures and Tables

**Figure 1 antioxidants-12-00457-f001:**
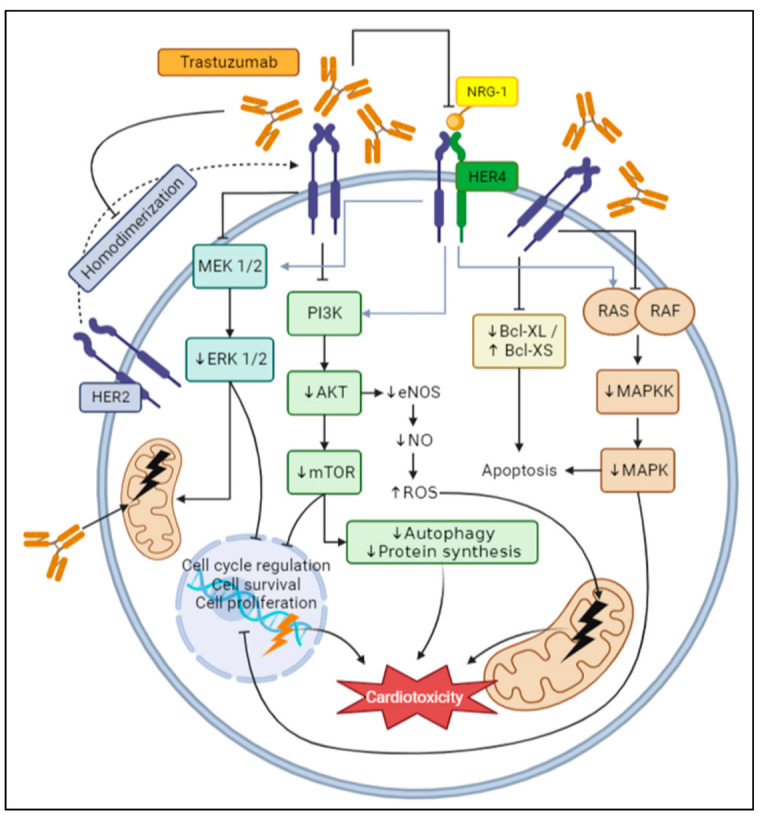
Molecular mechanisms of trastuzumab-induced oxidative stress in the cell. AKT, protein kinase B; Bcl-XL, B-cell lymphoma-extra-large; eNOS, endothelial nitric oxide synthase; ERK1/2, extracellular signal-regulated kinase 1/2; HER, human epidermal receptor; MAPK mitogen-activated protein kinase; MAPKK, MAPK kinase; MEK1/2 mitogen-activated protein kinase kinase 1/2; mTOR, mammalian target of rapamycin; NO, nitric oxide; NRG-1, neuregulin-1; PI3K, phosphoinositide 3-kinase; ROS; reactive oxygen species.

**Table 1 antioxidants-12-00457-t001:** Results of studies on the cardioprotective effects of antioxidants or drugs with antioxidant effects against trastuzumab toxicity. 1400W, N-(3-(aminomethyl) benzyl) acetamidine; AA, ascorbic acid; Ab, antibody; ALA, α-linolenic acid; BAX, BCL2-associated X protein; Bcl-XL; B-cell lymphoma-extra-large; CAT, catalase; CK-MB, creatine kinase MB; CoQ10; Coenzyme Q10; COX, cyclooxygenase; cTn-I, cardiac troponin I; DHA, docosahexaenoic acid; DOX, doxorubicin; EF, ejection fraction; FS, fractional shortening; GPX, glutathione peroxidase; GST, glutathione S-reductase; HER2, human epidermal growth factor receptor 2; IL-6, interleukin 6; i.p, intraperitoneally; LV, left ventricular; LVEF, LV ejection fraction; MDA, malondialdehyde; MEG, mercaptoethylguanidine; NAC, N-acetylcysteine; NACA, N-acetylcysteine Amide; Nf-kB, nuclear factor kappa-light-chain-enhancer of activated B cells; NT-proBNP, N-terminal pro b-type natriuretic peptide; NRVM, neonatal rat ventricular myocite; PARP, poly-ADP ribose polymerase; p.o., per os; PUFA, polyunsaturated fatty acid; ROS, reactive oxygen species; SDG, secoisolariciresinol diglucoside; SOD, superoxide dismutase; TNF-α, tumor necrosis factor alpha; TZB, trastuzumab.

*Antioxidant*	Model	Antioxidant Scheme	TZB or DOX Scheme	Results	Ref.
TZB as monotherapy					
AA	Cell line MCF10A and breast cancer cell lines MDA-MB-231, MCF-7, and SK-BR-3	5, 10, 15, and 20 mM during 2 h after TZB exposure	20 and 40 μg/mL during 24 h	AA significantly suppressed proliferation and decreased cell viability of breast cancer cell lines Additional inhibitory effect on the growth of breast cancer cell lines in combination with anticancer drugs	[79]
Allicin	Female Wistar albino rats (in vivo)	9 mg/kg/day p.o. for 5 weeks	6 mg/kg/week i.p. for 5 weeks	↑ SOD3, GPX1 and CAT activity ↓ Cardiac cell apoptosis	[80]
ALA and DHA (PUFAs, riched in omega-3)	BT-474 (HTB-20) cell line	ALA (50 and/or 100 μM) alone and combined with TZB DHA (50 and 100 μM) alone and combined with TZB In both experiments, treatment medium was refreshed after 48 h	10 μg/mL of TZB	↓ HER2-overexpressing breast cancer cell growth ↑ Apoptosis Modulates the HER2 signaling	[81]
Chrysin	Male Wistar rats (in vivo)	10 mg/kg/day i.p for 10 days	2.25 mg/kg/day i.p. injection for 10 days	Improvement of mitochondrial dysfunction ↓ TZB-induced pathological alterations	[82]
Coenzyme Q10 (nano emulsions)	Culture of human fetal cardiomyocytes (in vitro)	0.01, 0.1, 1, and 2.5% * w * /*v*	200 nM	↑ Cell viability ↓ ROS and leukotrienes production, lipid peroxidation, P65/NF-κB, interleukin-6, and 1β expression	[83]
Curcumin	Male Wistar rats (in vivo)	10 mg/kg/day i.p. for 10 days	2.25 mg/kg/day i.p. for 10 days	Improvement of mitochondrial dysfunction ↓ TZB-induced pathological alterations	[82]
Empagliflozin	Adult C57BL/6 mice (in vivo)	10 mg/kg, i.p. twice per week for 6 weeks	10 mg/kg/week i.p. for 6 weeks	TZB-induced cardiotoxicity was attenuated MDA levels normalized in presence of empagliflozin	[56]
Flavonoids from *Irvingia gabonensis*	Male Wistar albino rats (in vivo)	400 mg/kg/day p.o. for 7 days	2.25 mg/kg/day i.p. for 7 days	Mitigation of TZB attenuation of SOD, CAT, and GST activities Improvement in coronary artery histopathological alterations	[84]
NAC	Culture of neonatal rat cardiac ventricular myocytes (in vitro)	10 mM solution	1 g/mL	↑ Cardiomyocytes viability	[59]
Melatonin	Male Wistar rats (in vivo)	10 mg/kg/day p.o. for 7 days	4 mg/kg/day i.p. for 7 days	Attenuated the ↓ LV function and the impaired cardiac autonomic balance ↓ Elevation of serum cTn-I NT and proBNP levels, serum and cardiac MDA levels, and cardiac TNF-α and IL-6 levels. ↓ Cardiac apoptosis and autophagy disturbance Restored cardiac mitochondrial function and dynamics	[85]
Metformin	Male Wistar rats (in vivo)	250 mg/kg/day p.o. for 7 days	4 mg/kg/day i.p. for 7 days	Attenuated the ↓ LV function and the impaired cardiac autonomic balance ↓ E levation of serum cTn-I and NT-proBNP levels, serum and cardiac MDA levels, and cardiac TNF-α and IL-6 levels. ↓ C ardiac apoptosis and autophagy disturbance Restore cardiac mitochondrial function and dynamics	[85]
Polyphenols extracted from *Clerodendrum volubile*	Male Wistar albino Rats (in vivo)	400 mg/kg/day p.o. for 7 days	2.25 mg/kg/day i.p. for 7 days	Attenuated the ↓ of SOD, CAT, and GST activities Improvement coronary artery histopathological alterations	[84]
Ranolazine	C57Bl/6 mice (in vivo and ex vivo)	305 mg/kg/day p.o. for 10 days	2.25 mg/kg i.p. for 7 days	Attenuated ↓ of FS and EF ↓ TZB-induced apoptosis ↓ ROS production in NRVMs culture incubated with TZB plus isoproterenol	[86]
Selenium	White New Zealand rabbits	0.48 mg/kg diet content p.o.	6 or 8 mg/kg s.q .	No significant change of LVEF in the supplemented group Mitigation of mitochondrial changes	[87]
Thymoquinone	Male Wistar rats	0.5 mg/kg i.p. for 10 days	2.25 mg/kg/day i.p. for 10 days	Improvement of mitochondrial dysfunction ↓ TZB-induced pathological alterations	[82]
TZB combined with chemotherapy					
1400W	Male Sprague-Dawley rats (in vivo)	10 mg/kg/12 h i.p. 36 h before and continued for 72 h after DOX and TZB administration	10 mg/kg of TZB in combination with 20 mg/kg DOX i.p.	↓ Serum CK-MB and MDA levels ↑ SOD and GPX activities ↓ Interstitial edema, disorganization of the muscle fibers, and vacuolization	[88]
ALA	Wildtype C57BL/6 female mice (in vivo)	Daily prophylactic dietary regimen of 4.4% for 6 weeks	8 mg/kg of DOX and 3 mg/kg of TZB i.p. weekly for 3 last weeks after antioxidant treatment	Attenuated the ↓ of LVEF ↓ Nf-kB expression Attenuated the ↑ in BAX/Bcl-XL ratio and PARP expression Prevented the ↑ of cardiac mitochondrial dysfunction marker BNIP3 ↓ COX-derived oxylipins concentrations ↓ 8,9-dihydroxyeicosatrienoic acid concentration	[89]
Dexrazoxane	F344 rats	40 mg/kg before each TZB administration	0.8 mg/kg of DOX, once a week, for 2 weeks; then injection of 2 mg/kg TZB, once a week, for 2 weeks	Dexrazoxane pretreatment largely counteracted the severe damagesinduced by DOX + TZB usage Mitigation of the decrease in calpain-2 expression in the DOX + TZB group ↑ 1-month survival in the dexrazoxane pretreatment group	[90]
Flaxseed	Wildtype C57BL/6 female mice (in vivo)	Daily prophylactic dietary regimen 10% for 6 weeks	8 mg/kg of DOX and 3 mg/kg i.p. of TZB weekly for 3 last weeks after antioxidant treatment	Attenuated the decrease in the LVEF Preserved myofibril integrity ↓ Nf-kB expression Attenuated the increase in BAX/Bcl-XL ratio and PARP expression Prevented the increase of cardiac mitochondrial dysfunction marker BNIP3 ↓ COX-derived oxylipins concentrations ↓ 8,9-dihydroxyeicosatrienoic acid concentration	[89]
MEG	Male Sprague-Dawley rats (in vivo)	10 mg/kg/12 h i.p. 36 h before and continued for 72 h after DOX and TZB administration	10 mg/kg i.p. of TZB in combination with 20 mg/kg DOX	↓ Serum CK-MB and MDA levels ↑ Antioxidant enzyme activities (SOD and GPX) ↓ Interstitial edema, disorganization of the muscle fibers, and vacuolization.	[88]
Melatonin	Male Sprague-Dawley rats (in vivo)	10 mg/kg/12 h p.o. 36 h before and continued for 72 h after DOX and TZB administration.	10 mg/kg i.p. of TZB in combination with 20 mg/kg DOX	↓ Serum CK-MB and MDA levels ↑ Antioxidant enzyme activities (SOD and GPX) ↓ Interstitial edema, disorganization of the muscle fibers, and vacuolization.	[88]
NACA	C57Bl/6 female mice	250 mg/kg i.p. 30 min prior experimental treatment	10 mg/kg i.p. of TZB or 20 mg/kg i.p. DOX + 10 mg/kg i.p. TZB on one dose	Mitigated cardiovascular remodeling induced by TZB + DOX, by decreasing oxidative stress and cardiac cell apoptosis	[91]
Probucol	Wild type C57Bl/6 mice	15 mg/kg i.p. 2 weeks prior experimental treatment	10 mg/kg i.p. of TZB or 20 mg/kg i.p. of DOX + 10 mg/kg i.p. of TZB on one dose	↑ 10-day survival in mice with probucol pretreatment experimental groups. Mitigated echocardiographic impairment ↓ Cardiac cell apoptosis	[92]
SDG	Wildtype C57BL/6 female mice (in vivo)	Daily prophylactic dietary regimen 0.44% for 6 weeks	8 mg/kg DOX and 3 mg/kg TZB i.p. weekly for 3 last weeks after antioxidant treatment	Attenuated the decrease in the LVEF ↓ Nf-kB expression Attenuated the increase in BAX/Bcl-XL ratio and PARP expression. Prevented the increase of cardiac mitochondrial dysfunction marker BNIP3 No significant change concentration of COX-derived oxylipins ↓ 8,9-dihydroxyeicosatrienoic acid concentration	[89]

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
