# Peer review of "Antioxidant Protection against Trastuzumab Cardiotoxicity in Breast Cancer Therapy"

_antioxidants, 2023, doi:10.3390/antiox12020457_

Round 1
Reviewer 1 Report
This review explores the potential use of antioxidants as prevention of the cardiotoxicity mediated by Trasuzumab. In general it is well organized, but I think it would be improved for not expert readers on the subject if:
1.- A section on enzymatic and non-enzymatic action mechanisms of antioxidants is added (hopefully adding a scheme or table). It would also be good to add their regulatory mechanisms.
2.- In section 4, disaggregate the action mechanims of the different enzymatic and non-enzymes antioxidants molecules.
Other questions:
1.- What happens with the different oxidizing molecules during the use of Trasuzumab? Everyone increases, or only some, in this context, which happens with the levels of H2O2, no, lipid peroxides (4-hne), etc.
2.- The use of PUFAs has been suggested as a preventive therapy against otoxicity mediated by ROS. Will the same treatment for cardiotoxicity be possible?
Author Response
Comment 1: A section on enzymatic and non-enzymatic action mechanisms of antioxidants is added (hopefully adding a scheme or table). It would also be good to add their regulatory mechanisms.
Response: We did not understand this commentary, so we did not modify the article regarding this. We are completely open to a more detailed comment on your request if you feel it will positively impact the article.
Comment 2: In section 4, disaggregate the action mechanims of the different enzymatic and non-enzymes antioxidants molecules.
Response: Section 4 was divided according to the different antioxidants and complementary therapeutic agents.
Question 1: What happens with the different oxidizing molecules during the use of Trasuzumab? Everyone increases, or only some, in this context, which happens with the levels of H2O2, no, lipid peroxides (4-hne), etc.
Response: We found several articles citing a generalized increase in Reactive Oxidant Species, without differentiating which ones, for this reason we decided to include some of these articles to highlight the fact that oxidizing molecules in general increase in a context of antierbB2. However, in terms of specific oxidizing molecules, an increase in MDA has been described in patients with breast cancer who received trastuzumab as therapy (Dirican et al., 2014). This is written in section 4 of the manuscript.
Reviewer 2 Report
This review discusses the cardiotoxic effect of Trastuzumab in breast cancer, which is a much discussed and still unresolved topic. The authors argue, with appropriate references, that the use of antioxidants (a protective agent) may serve as an adjuvant therapy to Trastuzumab in breast cancer to prevent or decrese its cardiac toxicity.
This review provides an overview of current knowledge and stimulates researchers to continue addressing this issue to prevent or mitigate the known cardiotoxicity of the trastuzumab in breast cancer. The topic is interesting and important, and the paper is basically well written. Therefore, I think that this paper merits publication.
Suggestion
Table 1- Results of studies on the cardioprotective effect of antioxdants or drug with antioxdants effects against tastuzumab toxicity:
I suggest that the authors carefully check this table. For example, in the column “Antioxidant dose”, repeating the name of the drug is redundant.
Pag 4, line157, remove the name of the ref 39
Pag 4, line 177, remove the name of the ref 48
Pag 4, line 186, “GPX” instead “glutathione peroxidase”. Please check the acronym/abbreviations. Maybe the authors could add a list of abbrevations.
Pag 4, line 192, “organelle” instead of “organel”
Pag 5, line 226, “MAPK” instead “MAP
Please check the references according to the “Instruction for the Author”
Author Response
Comment 1: Table 1- Results of studies on the cardioprotective effect of antioxidants or drug with antioxidants effects against trastuzumab toxicity:
I suggest that the authors carefully check this table. For example, in the column “Antioxidant dose”, repeating the name of the drug is redundant.
Response: We checked the table and made some adjustments so that now the table is less redundant.
Comment 2: Pag 4, line157, remove the name of the ref 39
Response: The name of the ref 39 was removed.
Comment 3: Pag 4, line 177, remove the name of the ref 48
Response: The name of the ref 48 was removed.
Comment 4: Pag 4, line 186, “GPX '' instead “glutathione peroxidase”. Please check the acronym/abbreviations. Maybe the authors could add a list of abbreviations.
Response: “glutathione peroxidase” in line 186 has been modified to GPX. An abbreviation list has been added.
Comment 5: Pag 4, line 192, “organelle” instead of “organel”
Response: The word “organel” was corrected.
Comment 6: Pag 5, line 226, “MAPK” instead “MAP
Response: The abbreviation was corrected.
Comment 7: Please check the references according to the “Instruction for the Author”
Response: The journal of reference 96 and the title of reference 40 were modified. In addition, now all the years in the references are written in bold. The name of the journal was changed to its abbreviated version in the reference 7, 8, 10,12, 18,19, 20, 21, 33, 34, 38, 39, 42, 43, 49, 54, 60, 90
Reviewer 3 Report
This cardio-oncology review is important because it concerns the
use of antioxidants with trastuzumab in breast cancer. The authors
summarize several major studies of adjuvant therapy with trastuzumab
in experimental models and discuss their findings. Overall, this
review delivers significant considerations on breast cancer
therapy safety.
Author Response
The authors are thankful for your comment on the article.
Round 2
Reviewer 1 Report
Despite that you do not answered my comment 1. I think the current version is appropriate to be published.